# Moderate aerobic exercise, but not anticipation of exercise, improves cognitive control

**Maximilian Bergelt**[1], **Vanessa Fung Yuan**[1,2], **Richard O'Brien**[1], **Laura E. Middleton**[1]*, **Wellington Martins dos Santos**[3]

1 Department of Kinesiology, University of Waterloo, Waterloo, ON, Canada, 2 Department of Kinesiology, Brock University, St. Catharines, ON, Canada, 3 Department of Physical Education, University of Campinas, Campinas, São Paulo, Brazil

⊕ These authors contributed equally to this work.

* laura.middleton@uwaterloo.ca

## Abstract

### Background

Evidence suggests a single bout of exercise can improve cognitive control. However, many studies only include assessments after exercise. It is unclear whether exercise changes as a result, or in anticipation, of exercise.

### Objective

To examine changes in cognitive control due to moderate aerobic exercise, and anticipation of such exercise.

### Methods

Thirty-one young healthy adults (mean age 22 years; 55% women) completed three conditions (randomized order): 1) exercise (participants anticipated and completed exercise); 2) anticipation (participants anticipated exercise but completed rest); and 3) rest (participants anticipated and completed rest). Cognitive control was assessed with a modified Flanker task at three timepoints: (1) early (20 min pre-intervention, pre-reveal in anticipation session); (2) pre-intervention (after reveal); and (3) post-intervention. An accuracy-weighted response time ($RT_{LISAS}$) was the primary outcome, analyzed with a linear mixed effects modeling approach.

### Results

There was an interaction between condition and time (p = 0.003) and between session and time (p = 0.015). $RT_{LISAS}$ was better post-exercise than post-rest and post-deception, but was similar across conditions at other timepoints. $RT_{LISAS}$ improved across time in session 1 and session 2, but did not improve over time in session 3. There were also main effects of condition (p = 0.024), session (p = 0.005), time (p<0.001), and congruency (p<0.001).

**Data Availability Statement:** Deidentified data sufficient to reproduce all analyses can be found at https://osf.io/kx4mz/.

**Funding:** The author(s) received no specific funding for this work.

**Competing interests:** The authors have declared that no competing interests exist.

## Conclusions

Cognitive control improved after moderate aerobic exercise, but not in anticipation of exercise. Improvements on a Flanker task were also observed across sessions and time, indicative of a learning effect that should be considered in study design and analyses.

## 1 Introduction

Cognitive control is critical across the life span as it contributes to all aspects of daily life, including occupational, functional, and social activities. Research to improve cognitive control has generally focused on early life (as part of academic performance) and older adults (to reduce age-associated cognitive decline) [1–3]. Understanding of interventions to improve cognitive functions across the life course could impact rates of late-life dementia, by providing additional cognitive reserve capacity before function is affected.

A growing body research has examined the influence of physical exercise on cognitive control [4] with exercise now being considered a viable low cost option for maintaining neurological health as we age [5]. Optimistically, habitual physical activity (as reported by questionnaires) and aerobic fitness (as measured by maximal exercise tests) are both associated with better cognitive control [6–8]. Moreover, cognitive improvements and brain changes including increase in volume in areas of the brain (e.g., hippocampus) and connectivity of some brain networks can be observed after exercise interventions [4,5]. Though evidence from clinical trials of exercise is generally more inconsistent than observational studies [7,9–11], the most inclusive and most recent meta-analyses of randomized clinical trials generally conclude that exercise training (including aerobic, resistance, or mind-body exercise) is associated with better cognitive outcomes, though there is not yet support for prevention of dementia from these trials [12–19].

Improvements in cognitive control can also be observed after a single session of aerobic exercise [20,21]. Though recent meta-analyses indicate that acute exercise elicits a small positive effect on cognitive control [20,21], there were a number of factors that moderated the magnitude of the effects [20,21]. Potential moderators may include, but are not limited to, cognitive domain and task, exercise intensity and duration, and experimental design [20].

The cognitive benefits of exercise have been observed most consistently for executive functions. Executive functions refer to higher order cognitive processes that control basic cognitive processes for the purpose of goal-directed actions, including our ability to shift, inhibit, or update during a task [22–24]. Cognitive (or executive) control describes brain processes that guide goal-directed thoughts and behaviour. Cognitive control refers to the ability to focus on goal-related information and ignore irrelevant information and inhibit automatic responses. Cognitive control can be assessed using a modified Eriksen flanker task [25,26]. A number of studies have used a Flanker task to assess changes in cognitive control with exercise [5,6,27–29]. Arousal levels may influence cognitive performance across several cognitive domains, with evidence for changes in attention [30], working memory [31,32], and long-term memory [33–35]. In turn, arousal may change both due to exercise, as well as the anticipation of exercise; in turn, these changes in arousal may influence cognitive control. Anticipation of exercise, less studied in relation to cognitive control, is accompanied by a rise in ventilation [36] and increases in plasma cortisol and norepinephrine levels in some individuals [37]. Blood pressure and systemic vascular resistance were also observed to increase prior to a handgrip test, indicating sympathetic nervous system activation [38]. These studies suggest an increased

arousal prior to exercise, which could improve cognitive control pre-exercise [28,39], and at least partially account for the cognitive benefits observed with exercise.

Some study designs examining the effects of exercise on cognitive control would be vulnerable to confounding by pre-exercise arousal. Most acute studies assessing changes in executive functions have not included a pre-exercise assessment so would be unable to separate the cognitive effects of physical exercise from those related to the anticipation of exercise [29,40–45]. Even when pre and post-assessments were included, many existing studies failed to blind participants to the order of the sessions [29,40–45]. As a result, pre-intervention arousal might be different before the exercise intervention than before control sessions, confounding results.

The objective of this study was to examine the influence of the anticipation of exercise and physical exercise on Flanker task performance. We hypothesized that the anticipation of exercise and exercise itself would both improve cognitive control, as reflected by improved performance on the Flanker task. Additionally, we expected that the effect of exercise itself would be larger than the effect of anticipating exercise. To overcome the limitations of previous studies, this study used a deception condition to separate the anticipation of exercise from physical exercise itself. In addition, we include both pre- and post-intervention cognitive assessments (Flanker task).

## 2 Materials and methods

### 2.1 Participants

Thirty-one young healthy adults (aged 18–35 years) were recruited to the study by email and via posters placed around the University of Waterloo campus. Recruitment was active from July 2018 to April 2019. Participants were screened with the Physical Activity Readiness Questionnaire (PAR-Q+) to ensure safety to exercise [46]. Participants were excluded if they had: a history of heart disease (heart attack or operation, heart murmur, coronary artery disease, congenital heart disease, pacemaker); uncontrolled or hypertension; drop in blood pressure when rising from a seated position; neurological conditions (e.g. stroke, epilepsy, Parkinson's disease, or dementia); were taking beta blockers, anticoagulants, or anticholinergics; had chronic obstructive pulmonary disease; or had musculoskeletal impairments that cause more pain during exercise than tolerable. This study was approved by a University of Waterloo research ethics committee (ORE# 31106). All participants provided written informed consent.

### 2.2 Study design

This study used a repeated measures design to examine the influence of three conditions (exercise, exercise anticipation, and rest) on cognitive control. In the exercise anticipation session, participants came to the session expecting to exercise, but the researchers revealed they would instead rest after the first cognitive assessment. In the exercise session, participants expected to and did exercise. In the rest condition, participants expected to and did rest. Sessions occurred at least a week apart to lessen learning effects. The order of these conditions was counterbalanced. All participants were asked to refrain from caffeine 4 hours prior to the beginning of all study sessions.

### 2.3 Experimental protocol

In the first session, participants reported demographics and medical history, and completed the International Physical Activity Questionnaire (IPAQ) [47]. In all sessions, participants then started the experimental protocol with a practice block of 50 trials of a modified Flanker task (described below). Subsequently, cognitive control was assessed with 200 trials of the

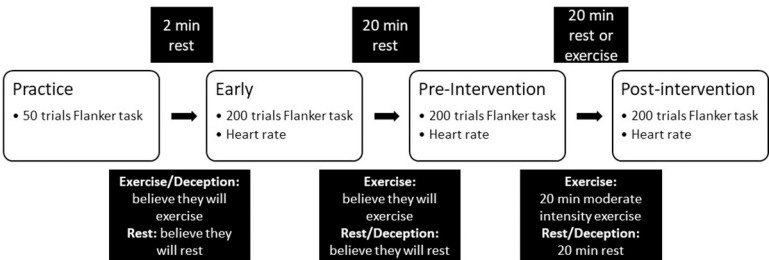

**Fig 1. Experimental design.** The three conditions (exercise, rest, and anticipation) were in randomized order.

modified Flanker task at three timepoints: (1) early (pre-reveal in the exercise anticipation condition); (2) pre-intervention (post-reveal in the exercise anticipation condition, but pre-exercise/rest; 20 min after the early assessment); and (3) post-intervention. The experimental protocol is shown in Fig 1.

During the exercise session, participants performed approximately 3 minutes of acclimatization on the bike followed by 20min of moderate intensity aerobic exercise on a stationary cycle ergometer and 3 minutes of self-paced cool down. Participants were instructed to maintain a cadence of 90±5 rpm. Cycle ergometer resistance was adjusted to maintain a rating of perceived exertion (RPE) of 13 measured on the Borg scale of perceived exertion (6–20 version) [48], consistent with moderate intensity exercise. RPE and resistance (watts) were recorded every 2 minutes. In both the exercise anticipation and rest sessions, participants completed 26 minutes of quiet rest on the stationary cycle ergometer, with only the anticipation of exercise (and reveal) varying between sessions. Heart rate was recorded every 2 minutes during all interventions using a Polar heart rate monitor.

## 2.4 Measures

The primary outcome was a modified Eriksen Flanker task [25], used to probe the evaluation of cognitive control. The modified Flanker Task consisted of five arrow heads displayed on the screen. The participant was asked to respond to the direction of the centre target arrow by pressing one button if it pointed left and another if it pointed right. The flanking arrows could point in the same direction as the centre arrow (congruent condition, e.g. >>>>>) or in the opposite direction from the centre arrow (incongruent condition, e.g. <<><<). There was an even distribution of congruent and incongruent trials within each 200-trial block. The modified Flanker Task was created and delivered using STIM2 software (Compumedics Neuroscan, El Paso, TX, USA).

The task was performed in a small room with a wall and a divider to prevent visual distraction. Participants were instructed to look at a small white fixation-cross in the middle of a black screen where the target stimuli appeared and to respond as quickly as possible to the stimulus when it appeared. The response pad was placed on a table centered allowing for an elbow's length of reach. Participants responded with their left index for arrows point left and their right index for arrows pointing right. Participants were seated 120 cm away from a 24-inch computer monitor. The center of the stimulus was 27 cm from the surface of the table and had a height of approximately 10 cm. Each stimulus was displayed for 150ms with a 1000ms response window. There was a 1250 or 1750 (randomized) ms inter-trial duration. A minimum response time of 200ms was required for correct responses to eliminate anticipatory responses.

Accuracy and response time were collected by the Stim2 software. Response time was only considered for correct trials. Since accuracy varied by condition and time, modified Flanker

task performance was quantified by an accuracy and variability adjusted response time score, the Linear Integrated Speed Accuracy Score (referred to as $RT_{LISAS}$) [49,50]. We chose to report results using $RT_{LISAS}$ (as opposed to the more typical RT) to account for the observed speed-accuracy tradeoffs made by the participants. Though still a relatively new measure, $RT_{LISAS}$ has already been used in a number of studies of cognition over the past 3 years, see for example [51–54]. The $RT_{LISAS}$ score was calculated using the following equation:

$$RT_{LISAS} = RT + \left( \frac{sd\ RT}{sd\ PE} \times PE \right)$$

RT and PE are the participants' mean response time and proportion error (1-accuracy) for each block of flanker responses, separated by congruency. The terms sd RT and sd PE refer to the participants overall standard deviation in response time and proportion error for each block of flanker responses, again separated by congruency. If PE was zero (100% accuracy), the latter portion of the equation was set to zero. In cases of high accuracy, the changes observed in $RT_{LISAS}$ were more influenced by response time than by accuracy. In these cases, $RT_{LISAS}$ can be considered mostly as an accuracy adjusted response time measure.

The interpretation of $RT_{LISAS}$ is analogous to the interpretation of response time; namely, lower scores are better. This is because as PE decreases (i.e. accuracy increases), the later part of the equation shrinks towards 0. If PE is equal to zero (100% accuracy), then the measure is simply the response time.

## 2.5 Statistical analysis

All analyses were conducted in R v.3.6.0 [55]. Differences in $RT_{LISAS}$ were analyzed using linear mixed effects models built using the lme4 R package v1.1–21 [56], with significance tests provided by the lmerTest R package v3.1–0 [57]. Fixed effects of condition (exercise, exercise anticipation, rest), time (early, pre-intervention, post-intervention), congruency (congruent, incongruent), and session (first, second, third) were included. The rate of no response and gender were included as covariates. Session, condition, time, and congruency where fully interacted. Statistical significance of fixed effects was determined using Satterthwaite Type III ANOVA tests conducted on the linear mixed effects models using the ANOVA function in R. Significance for all analyses was defined as $p<0.05$. Data driven post-hoc contrasts were conducted using the emmeans function from the emmeans R package v1.4.1 [58]. All contrasts used the Tukey correction for multiple comparisons and the Satterthwaite method for calculating degrees of freedom. Pairwise effect sizes (as standardized mean differences, SMDs, also known as Cohen's d) were obtained with the emmeans package. The SMDs were calculated using the pairwise differences of the model estimates, divided by the estimated population standard deviation as obtained from the residual standard deviation of the model. Effect sizes were classified according to general convention with .2 as small, .5 as medium. and .8 as large.

## 3 Results

### 3.1 Participant characteristics

Participants had an average age of 22.0 years (standard deviation, sd = 0.90, range: 20–25) and 55% [17] were female. Average activity level as measured by the IPAQ was 2797 MET-min/wk (sd = 1898, range: 360–6558). Flanker data from all 31 participants was included in the analyses though 7 participants (3 male, 4 female) were missing a proportion of their data (proportion missing: 11% to 67%).

**Table 1. Exercise characteristics by condition and time.**

| Time | Rest | Deception | Exercise | P-Value |
|---|---|---|---|---|
| Early HR | 78 (11) | 81 (11) | 81 (12) | 0.287 |
| Pre-Intervention HR | 78 (10) | 79 (11) | 78 (11) | 0.784 |
| During Intervention HR* | 75 (9) | 76 (9) | 132 (20) | < .001 |
| Post-Intervention HR | 76 (9) | 77 (12) | 91 (17) | < .001 |
| During Intervention RPE† | - | - | 13 ± 1 | - |
| During Intervention Watts† | - | - | 66 ± 29 | - |

All values are represented by mean ± SD unless otherwise specified.

* Mean of the median values of the ten time points collected two minutes apart during the intervention

† Queried only during the exercise condition

## 3.2 Exercise characteristics

Characteristics of participants during exercise are displayed in Table 1. The percent of maximum heart rate was similar across all three sessions at early and pre-intervention times. However, the percent of maximum heart rate attained by participants during the intervention and post-intervention was significantly higher in the exercise session than in the resting and anticipation sessions (during intervention: 67% ± 10% for exercise versus 38% ± 5%. for rest and 39% ± 5% for deception; post-intervention: 46% ± 9% for exercise versus 38% ± 5% for rest and 39% ± 6% for deception). The average recorded RPE during exercise was 13 (sd = 1), which aligned with a moderate intensity as intended. The average mechanical power of participants during exercise was 66 watts (sd = 29).

## 3.3 Flanker results

There was a significant interaction effect of condition x time for $RT_{LISAS}$ ($F(4,419.86) = 4.13$, $p = 0.003$). The $RT_{LISAS}$ for the rest, deception, and exercise conditions was not significantly different at the early assessment (rest: M = 368, SEM = 8.57; deception: M = 367, SEM = 8.59; exercise: M = 367, SEM = 8.57; all pairwise p = 1.000; SMDs: 0.01–0.04) or the pre-intervention assessment (rest: M = 360, SEM = 8.58; deception: M = 355, SEM = 8.59; exercise: M = 357, SEM = 8.54; all pairwise p>0.95; SMDs: 0.08–0.21). However, in the post-assessment, the $RT_{LISAS}$ in the exercise condition (M = 336, SEM = 8.54) was significantly lower when compared to both the rest condition (M = 352, SEM = 8.57, t = 3.38, df = 422, p = 0.022, SMD: 0.66) and deception condition (M = 358, SEM = 8.62, t = 4.62, df = 421, p<0.001, SMD: 0.91). The rest and deception conditions were not significantly different at post-assessment (t = 1.28, df = 420, p = 0.937, SMD: 0.25). $RT_{LISAS}$ by condition and time is presented in Fig 2.

There was also significant interaction of session x time for $RT_{LISAS}$ ($F(4,419.89) = 3.12$, $p = 0.015$). In the first session (regardless of session type), the $RT_{LISAS}$ decreased (marginally significantly) from early (M = 377, SEM = 8.54) to pre-intervention (M = 364, SEM = 8.49, t = 3.08, df = 420, p = 0.057, SMD: 0.57) and from pre-intervention to post-intervention (M = 349, SEM = 8.51, t = 3.46, df = 420, p = 0.017, SMD: 0.63). In the second session, $RT_{LISAS}$ decreased from early (M = 366, SEM = 8.56) to pre-intervention (M = 351, SEM = 8.58, t = 3.15, df = 420, p = 0.045, SMD: 0.90) but not from pre-intervention to post-intervention (M = 344, SEM = 8.56, t = 1.55, df = 420, p = 0.831, SMD: 0.30). In the third session, $RT_{LISAS}$ showed no significant differences between early (M = 360, SEM = 8.66) and pre-intervention (M = 356, SEM = 8.66, t = 0.80, df = 420, p = 0.997, SMD: 0.16) or pre-intervention and post-

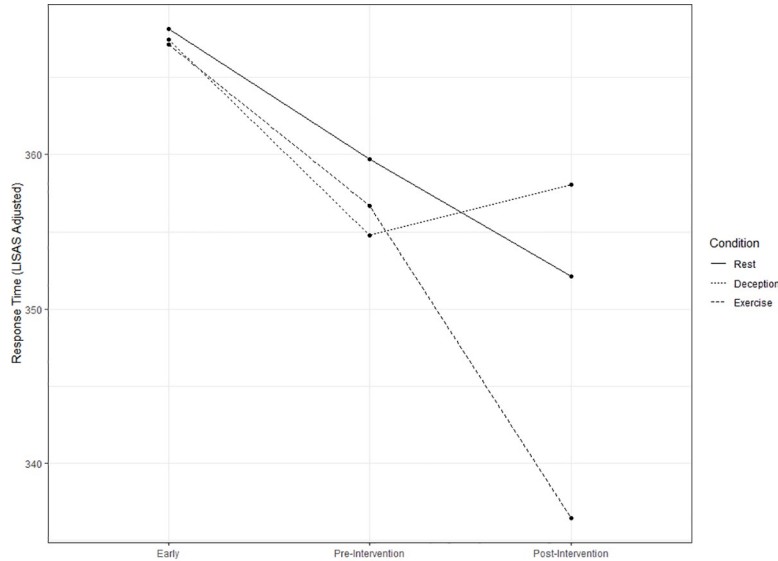

**Fig 2. Condition by time plot.** Points represent the model estimated accuracy adjusted response time ($RT_{LISAS}$) for each level of condition (rest, deception, exercise) and time (early, pre-Intervention, post-intervention).

intervention (M = 354, SEM = 8.66, t = 0.51, df = 420, p = 1.000, SMD: 0.10). $RT_{LISAS}$ by session and time is presented in Fig 3.

There were also main effects of condition (F(2,423.88) = 3.78, p = 0.024), time (F(2,419.91) = 25.05, p<0.001), session (F(2,425.03) = 5.47, p = 0.005), and congruency (F(1,419.82 = 226.14, p<0.001). $RT_{LISAS}$ was significantly higher in the rest condition (M = 360, SEM = 8.16) compared to the exercise condition (M = 353, SEM = 8.14, t = 2.36, df = 426, p = 0.050, SMD: 0.28) and the deception condition (M = 360, SEM = 8.16) compared to the exercise condition (t = 2.42, df = 424, p = 0.042, SMD: 0.28). The difference between the rest and deception

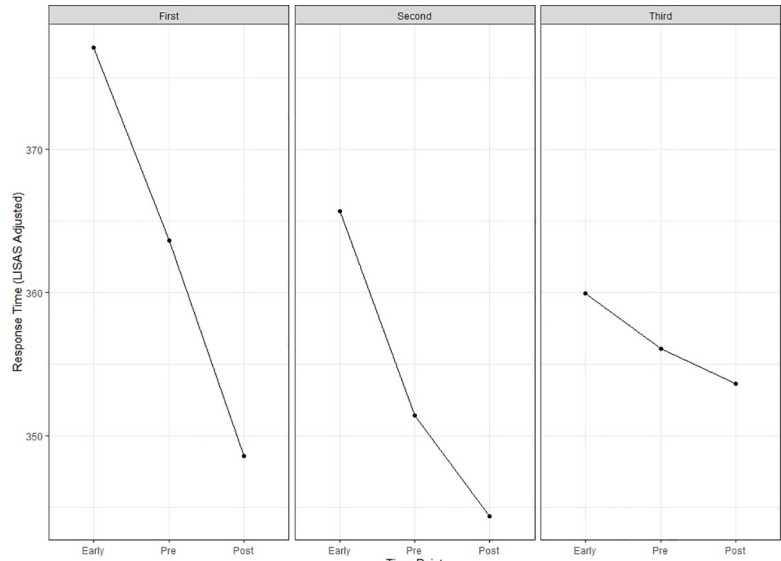

**Fig 3. Session by time plot.** Points represent the model estimated accuracy adjusted response time for each level of session (first, second, third) and time (early, pre-intervention, post-intervention).

conditions was not significant (t = 0.04, df = 422, p = 1.000, SMD: 0.10). $RT_{LISAS}$ was also higher at time 1 (M = 368, SEM = 8.14) than time 2 (M = 357, SEM = 8.14, t = 3.98, df = 420, p<0.001, SMD: 0.44) and $RT_{LISAS}$ at time 2 was higher than time 3 (M = 349, SEM = 8.14, t = 3.10, df = 420, p = 0.006, SMD: 0.34). $RT_{LISAS}$ was also higher in the first session (M = 363, SEM = 8.13) compared to the second (M = 354, SEM = 8.16, t = 3.26, df = 428, p = 0.003, SMD: 0.39) but not the third session (M = 357, SEM = 8.20, t = 2.22, df = 427, p = 0.068, SMD: 0.27). The difference between the second and third sessions was not statistically significant (t = 1.00, df = 421, p = 0.578, SMD: 0.12). Finally, incongruent trials had higher $RT_{LISAS}$ (M = 374, SEM = 8.07) compared to congruent trials (M = 342, SEM = 8.07, t = 15.04, df = 420, p<0.001, SMD: 1.36).

## 4 Discussion

This study evaluated the impact of moderate aerobic exercise, and anticipation of exercise, on cognitive control. Our results confirmed that moderate aerobic exercise improves cognitive control, and that this benefit occurs independent of anticipation (and possible arousal) that occurs prior to exercise. In addition, secondary analyses indicated that there are significant learning effects with a modified Flanker task, both within and across sessions. Together, these results confirm aerobic exercise as part of a strategy to augment cognitive function and suggest that research using a Flanker task should carefully consider learning effects in the study design, target sample size, and analysis.

### 4.1 Effect of exercise and anticipation of exercise

Our study provides support for the acute cognitive benefits of exercise, by demonstrating positive changes in cognitive control with exercise (in line with our hypothesis) while addressing previous methodological shortcomings. Most prior studies examining exercise and cognitive control only included post-exercise assessments [29,40–45]. Even when pre and post-assessment measures were used, many existing studies failed to blind participants to the order of the sessions [29,40–45]. As a result, pre-intervention arousal might be different before the exercise intervention than before control sessions, thereby confounding results. As a result, in prior research, it was unclear whether cognitive control improved in anticipation of exercise or as a result of exercise. Both were reasonable given prior observations of sympathetic nervous system activation pre-exercise [36–38], and the known effects of sympathetic nervous system activation on cognitive control [28,39]. Here, we included two pre-exercise cognitive assessments and a deception condition to separate the anticipatory and physical contributions of exercise. In this study, Flanker task performance was better after exercise condition than after rest, regardless of whether participants had been anticipating exercise.

Our results counter our hypothesis that arousal associated with anticipating exercise would improve cognitive control [28,39]. In this study, Flanker task performance was similar across sessions at the early (pre-reveal) and pre-intervention times. Furthermore, Flanker task performance did not differ in the deception condition between when participants were anticipating exercise (early assessment) and when they knew they were resting (pre-intervention). There are two explanations for the lack of observed differences due to the anticipation of exercise: (1) the anticipation of exercise did not change arousal levels or (2) arousal caused by anticipating exercise did not alter cognitive control.

We hypothesized that cognitive control would improve with the anticipation of exercise since prior research indicated that several markers of arousal and sympathetic nervous system activation changed prior to exercise onset (e.g., ventilator rate, cortisol levels, blood pressure) [36–38]. However, another study noted that there was considerable individual variability in

the pre-exercise stress response and arousal changes [37]. As a result, it is reasonable to think that our participants did not experience pre-exercise arousal changes, or that changes were too small to induce cognitive improvements. Although the percent of heart rate maximum was 2 percentage points higher in the deception and exercise conditions than in the rest condition in this study, differences were not statistically significant. Any alteration in arousal due to the anticipation of exercise may have been too small to be detected by heart rate, a very coarse measure of arousal and sympathetic nervous system activation. However, it is impossible to confidently determine whether or not participants experienced arousal changes due to the anticipation of exercise in this study.

The second possibility is that a change in arousal was elicited by anticipation of exercise but that it was insufficient to, or simply did not, change cognitive control as measured by the flanker. Arguably, this is less likely as there is good evidence that arousal levels impact cognitive control. For example, a meta-analysis concluded that acute increases in cortisol enhanced cognitive control in the short term [39]. In addition, catecholamine neurotransmitters are believed to be involved in the regulation of arousal and cognitive control [59–61], though we did not measure these in this study. It seems likely that if the anticipation of exercise caused a significant change in arousal, it would have had an effect on the flanker task. However, it should be noted that our Flanker task had an even number of congruent and incongruent trials as opposed to more congruent than incongruent trials which has been shown to increase the Flanker congruency effect and make the task more cognitively demanding [62]. It is possible that had we adopted this uneven distribution of congruent and incongruent trials, the effect of the anticipation of exercise may have been stronger due to the heightened difficulty level of the task.

The observation that anticipation of exercise did not result in cognitive improvement, but exercise itself did, may point to a unique or additional mechanism of action for exercise that is dissociable from arousal. For example, it is possible that the improvement in cognitive control resulted at least partially from acute changes in brain derived neurotrophic factor (BDNF) or insulin-like growth factor-1 (IGF-1) activity, as BDNF [63] and IGF-1 [64] have both been associated with improved cognitive control but are not typically associated with psychologically induced arousal. However, levels of these growth factors were not measured in this study and so that hypothesis is merely speculative.

## 4.2 Learning effect

An incidental finding of this study was detection of a significant learning effect for the flanker task within and across sessions. In our perusal of the literature this appears to be poorly characterized. We found one study of the psychometric properties of a novel choice reaction time task that demonstrated learning effects across repeated testing [65]; one study that found a learning effect of a choice reaction time task from the first to the second day [66], though not across time points on the first day; and one study that reported flanker task improvement from an initial to a final session [67], though it was not certain whether this was due to a test-retest learning effect or a result of the interceding tasks that the participants performed. Our results support these limited findings and further characterize the learning effect of the flanker task. We found that performance improves both across sessions as well as across time points during the same session, in particular on the first day.

The methodological implications of this substantial learning effect are broad. Very often when using psychometric tests, particularly choice reaction time tests, papers (including ours) state that training was given to participants before administering the test in order to eliminate learning effects from the data [66]. However, it is not often demonstrated that this training

actually leads to a plateau in performance; in our case, it was demonstrable that our training block of 50 Flanker trials performed at the start of each session absolutely did not do this. In our case, this was not a huge issue as we had counterbalanced our conditions meaning that our results were not biased by the learning effect. However, there may well be situations in which researchers judge it difficult to use strategies such as counterbalancing or control groups that would address this issue and may subsequently justify not doing those things by leaning on the assumption that the task training given to participants rendered the learning effect negligible. Our results show that for the Flanker task this is likely to be an untenable assumption which is something that must be considered when designing studies using this task.

### 4.3 Limitations

The most significant limitation in this study is that physiological arousal was evaluated using a very coarse measure and therefore we cannot confidently know whether or not a stress response occurred with the anticipation of exercise. As a result, we cannot conclusively determine whether or not arousal occurred due to the anticipation of exercise. Other limitations include a relatively small sample size drawn from a homogenous group and the use of just one measure of cognitive function. Our sample was quite small at only 31 participants. This may have limited our ability to detect a small but true effect of exercise anticipation, However, our results indicate a very small anticipatory effect (0.03 between rest and deception and 0.04 between rest and exercise at time 1, necessitating a sample size over 3800 to detect). Our participants were all young healthy adults, comprised entirely of undergraduate and graduate university students, therefore limiting generalizability. It is not prudent to assume that the benefits seen in this study necessarily generalize to other age groups or clinical populations. Additionally, our only measure of cognitive function was a modified Flanker task. This task only probes attentional control and cognitive inhibition meaning that the improvements seen following exercise in this study may only apply to those aspects of cognitive control and not others, such as memory or cognitive flexibility.

### 4.4 Future directions

This study provides evidence for the benefits of exercise to cognitive function. Future studies should continue to probe these benefits are, who they occur to, and how they arise. We also provide evidence that cognitive function does not benefit from the anticipation of exercise, at least for measures of attentional control and cognitive inhibition. However, open questions still include whether anticipation generates a measurable stress response, for whom anticipation generates a stress response, what activities generate an anticipatory stress response, and whether the stress response benefits cognitive function in those in whom it occurs. Finally, our incidental finding of a learning effect across times and sessions for the modified Flanker task carries important methodological implications. Given that a learning effect is present, it is necessary to take this into account when designing future experiments, for example by counterbalancing conditions or including control groups. There appears to be a serious gap in the literature when it comes to characterizing the learning effect of the flanker, and so this too is an area that needs to be further explored.

### Supporting information

**S1 File. Summary results of all statistical analyses.**
(PDF)

## Author Contributions

**Conceptualization:** Maximilian Bergelt, Vanessa Fung Yuan, Richard O'Brien, Laura E. Middleton.

**Data curation:** Maximilian Bergelt.

**Formal analysis:** Maximilian Bergelt, Laura E. Middleton.

**Funding acquisition:** Laura E. Middleton.

**Investigation:** Maximilian Bergelt, Vanessa Fung Yuan, Wellington Martins dos Santos.

**Methodology:** Maximilian Bergelt, Vanessa Fung Yuan, Richard O'Brien, Laura E. Middleton.

**Project administration:** Maximilian Bergelt, Laura E. Middleton.

**Resources:** Laura E. Middleton.

**Software:** Maximilian Bergelt.

**Supervision:** Maximilian Bergelt, Laura E. Middleton.

**Validation:** Vanessa Fung Yuan.

**Visualization:** Maximilian Bergelt.

**Writing – original draft:** Maximilian Bergelt, Vanessa Fung Yuan, Richard O'Brien, Laura E. Middleton.

**Writing – review & editing:** Maximilian Bergelt, Vanessa Fung Yuan, Richard O'Brien, Laura E. Middleton, Wellington Martins dos Santos.

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
