## [Decision Letter · Decision Letter 0]

19 Aug 2020

PONE-D-20-21829

Moderate Aerobic Exercise, but not Anticipation of Exercise, Improves Cognitive Control

PLOS ONE

Dear Dr. Maximilian Bergelt,

Thank you for submitting your manuscript to PLOS ONE. After careful consideration, we feel that it has merit but does not fully meet PLOS ONE’s publication criteria as it currently stands. Therefore, we invite you to submit a revised version of the manuscript that addresses the points raised during the review process.

Two experts in the field evaluated the manuscript. Although they feel that the manuscript has merit, both of them have identified methodological issues. In particular, the second reviewer is concerned with the even distribution of congruent and incongruent trials in the modified Flanker task used. The academic editor agrees that this is a crucial issue that should be clarified by the authors. Moreover, please carefully address all comments made by each reviewer.

We look forward to receiving your revised manuscript.

Kind regards,

Samuel Penna Wanner, Ph.D.

Academic Editor

PLOS ONE

Additional Editor Comments:

In addition to the comments made by the two reviewers, please consider:

1. To run experiments to address if an even distribution of congruent and incongruent trials affects the modified Flanker task outcomes.

2. To present the heart rate and rating of perceiving exertion data (both the means and SD) without decimal places. Also, please describe how these two intensity parameters were measured (e.g., equipment and scale used).

3. To present the mechanical power output during the exercise trial.

4. To provide more details in the legend of figures, particularly of figures 2 and 3. Please remember that a figure must stand alone.

5. To replace “didn’t” with “did not” in lines 272 and 294.

6. To briefly indicate, in the Results section, whether the score used (i.e., RTLISAS) was more influenced by the response time or accuracy. In general, the literature suggests that an acute bout of physical exercise improves response time but does not change accuracy. Please also explain why a lower score means better performance (lower response time and/or lower accuracy?).

Journal Requirements:

2. Please provide further details regarding how participants were recruited, including the participant recruitment date.

Reviewers' comments:

Reviewer's Responses to Questions

**Comments to the Author**

1. Is the manuscript technically sound, and do the data support the conclusions?

Reviewer #1: Yes

Reviewer #2: Yes

2. Has the statistical analysis been performed appropriately and rigorously? 

Reviewer #1: Yes

Reviewer #2: Yes

3. Have the authors made all data underlying the findings in their manuscript fully available?

Reviewer #1: Yes

Reviewer #2: Yes

4. Is the manuscript presented in an intelligible fashion and written in standard English?

Reviewer #1: Yes

Reviewer #2: Yes

5. Review Comments to the Author

Reviewer #1: The concept of the study is good and interesting. Methodology could be more detailed. However, the results can be further explored as well as their relationship with the existing literature.

.Abstract needs some review. The experimental situations are not included as well the deception concept.

.Introduction is clear and concise.

Line #46 The authors needs to clarify if "brain changes" are better explored, would they be positive or negative?

.Methods

The cardiorrespiratory fitness or mechanical power of participants could be included to observe the fitness level of participantes.This is important considering that different effects of acute aerobic exercise on cognition are observed both at different intensities of exercise (as mentioned by the author in the introduction) as well as at different levels of physical fitness.

The sample size is not adequate. A posteriori sample size calculation is stimulated.

Effect size calculation is also stimulated to provide the magnitude of the main effects observed in the results.

Line #49: Considering that learning effect is an important topic of this study, the authors should explain or refer to a protocol already published in literature that tested if learning effect is minmized after only one week.

Line #153: It should be informed whether the use of several criteria to analyze the parameters of cognitive task is arbitrarily proposed by authors or is already in the literature. If so, please cite this original work.

.Results

The results are too much descriptive and data are repeated at some points data.

Line #184: Regarding missing data of some participants concerning cognitive task parameters, were the remmaining data considered for analysis or were the participants excluded? Was there any criteria for this decision?

Line #189: The data could be reported according to relative values (i.e. %HRmáx) to increase the individuality of measure. The proximity of anaerobic thresold influences their cognitive control response, for example.

.Discussion

The discussion is superficial. A very simple relationship is made with the findings of the literature.

An important finding, the "learning effect" is poorly explorated. The authors can explore and relate these data with literature in conclusion, not just in the future directions.

Line #291: Here the authors need to reinforce that catecholamines and cortisol were not assessed in the present study.

Lines #291 and 298: unpadronized citation. These citations do not follow the journal's instructions

Line #316: the term 'strong' appears to be inappropriately used. The authors reinforce multiple limitations of the present study. Also, they did not measure the magnitude of statistical difference (i.e. effect size).

Reviewer #2: The authors presented a well-written manuscript on the effects of moderate exercise, and its anticipation effect on cognitive control. The authors also used a relatively new measure of the Flanker test.

The results are interesting and bring novelty to the literature. However, some methodological problems were found.

Major comments:

The authors adopted an even distribution of congruent and incongruent trials. However, Lehle and Hubner [1] pointed out the need to adopt a high frequency of congruent stimulus compared with incongruent, once with an even distribution, the participants can adapt to them. In that way, we cannot know if anticipation exercise did not affect cognition due to the low difficulty level of the task.

The sample was composed of 55% of females. The literature indicates that cognitive control is modulated by estrogen levels [2-4]. However, the authors did not control participants’ menstrual cycle, and we did not know at which extend this can be influencing the results.

Minor comments:

Line 40: The authors introduce the paragraph talking about exercise benefits to cognition. Then, they talk about the physiological and metabolic benefits of exercise. I believe this second sentence is not necessary.

Line 110: It seems that the sentence has a typing error.

Line 130: The author affirmed that they recorded the resistance during exercise condition, but they did not report these data in the results section.

Table 1. It not common to use this grid format in tables. I believe, for future submissions, the authors should fix it.

1. Lehle C, Hubner R. On-the-fly adaptation of selectivity in the flanker task. Psychonomic Bulletin & Review 2008;15(4):814-8

2. Colzato LS, Hertsig G, van den Wildenberg WP, Hommel B. Estrogen modulates inhibitory control in healthy human females: evidence from the stop-signal paradigm. Neuroscience 2010;167(3):709-15 doi: 10.1016/j.neuroscience.2010.02.029[published Online First: Epub Date]|.

3. Hodgetts S, Weis S, Hausmann M. Estradiol-related variations in top-down and bottom-up processes of cerebral lateralization. Neuropsychology 2017;31(3):319-27 doi: 10.1037/neu0000338[published Online First: Epub Date]|.

4. Jacobs E, D'Esposito M. Estrogen shapes dopamine-dependent cognitive processes: implications for women's health. J Neurosci 2011;31(14):5286-93 doi: 10.1523/jneurosci.6394-10.2011[published Online First: Epub Date]|.

6. PLOS authors have the option to publish the peer review history of their article (what does this mean?). If published, this will include your full peer review and any attached files.

Reviewer #1: **Yes: **João Gabriel Silveira-Rodrigues

Reviewer #2: **Yes: **Larissa Oliveira Faria

---

## [Author Response · Author response to Decision Letter 0]

15 Oct 2020

Dear Dr. Samuel Penna Wanner,

Thank you for the opportunity to revise our manuscript as well as the thoughtful comments and suggestions. In our revised manuscript, we have carefully considered your and the reviewers’ suggestions and have updated the manuscript accordingly. In the rest of the letter we respond to each of the comments raised by you and the reviewers. Reviewer comments are above, while our responses are below, preceded by '>'. Within the manuscript new additions or changes that were made to satisfy reviewer concerns are highlighted in yellow and tracked changes is turned on. In addition to changes to address reviewer comments, we have also made changes to address some grammatical, spelling, and stylistic errors that we found in our own review of the manuscript. Overall, we feel that the quality of our paper is substantially improved thanks to the changes we have made to address the issues raised in this review.

Cheers,

Max Bergelt

Additional Editor Comments:

In addition to the comments made by the two reviewers, please consider:

1. To run experiments to address if an even distribution of congruent and incongruent trials affects the modified Flanker task outcomes.

> Most prior studies of exercise in relation to the Flanker task have used an even distribution of congruent and incongruent trials (recent examples: (1–5)), in line with our design. Unfortunately, in-person research is not possible at our University at this time so it is impossible to systematically examine whether a harder Flanker congruency distribution would have been more sensitive to anticipation of exercise. However, we had discussion regarding this point, and it will certainly be something we look into for any future studies we do in this area.

2. To present the heart rate and rating of perceiving exertion data (both the means and SD) without decimal places. Also, please describe how these two intensity parameters were measured (e.g., equipment and scale used).

>Done

3. To present the mechanical power output during the exercise trial.

>Done

4. To provide more details in the legend of figures, particularly of figures 2 and 3. Please remember that a figure must stand alone.

>Done

5. To replace “didn’t” with “did not” in lines 272 and 294.

>Done

6. To briefly indicate, in the Results section, whether the score used (i.e., RTLISAS) was more influenced by the response time or accuracy. In general, the literature suggests that an acute bout of physical exercise improves response time but does not change accuracy. Please also explain why a lower score means better performance (lower response time and/or lower accuracy?).

>We have added in an explanation for why a lower score means improved performance to the Methods section of our paper. In cases of high accuracy (as in this study), RTLISAS can be considered an accuracy-adjusted response time, so the primary driver of changes in RTLISAS is changes in reaction time. A study comparing methods to combine accuracy and RT considered RTLISAS as the method adding the most value (6). 

 

Journal Requirements:

>We have updated the manuscript and our files to ensure that titles, references, figures, and supplementary files are formatted and named correctly.

2. Please provide further details regarding how participants were recruited, including the participant recruitment date.

>We have added more detail as to how participants were recruited (by email and posters) and the dates recruitment was active (July 2018 to April 2019).

>Done.

 

Response to reviewers

Reviewer #1: The concept of the study is good and interesting. Methodology could be more detailed. However, the results can be further explored as well as their relationship with the existing literature.

.Abstract needs some review. The experimental situations are not included as well the deception concept.

>We have clarified the experimental design in the abstract, within the limitations of the word limit.

.Introduction is clear and concise.

>Thank you.

Line #46 The authors needs to clarify if "brain changes" are better explored, would they be positive or negative?

>This is a good point, ‘brain changes’ is vague. We have changed the wording specify increased volume of areas of the brain (e.g. hippocampus) and connectivity in some brain networks.

.Methods

The cardiorrespiratory fitness or mechanical power of participants could be included to observe the fitness level of participants. This is important considering that different effects of acute aerobic exercise on cognition are observed both at different intensities of exercise (as mentioned by the author in the introduction) as well as at different levels of physical fitness.

>Thank you for catching this. We did calculate the mechanical power of participants during the exercise session, though we overlooked including it in the manuscript. We have now added it to the section 3.2 Exercise Characteristics.

The sample size is not adequate. A posteriori sample size calculation is stimulated.

>We agree that the sample size is not adequate to confidently assert that exercise anticipation does not have an effect, and we have added a few sentences to the limitations section of our paper to underscore this for readers. Based on our effect sizes (acknowledging they may be misleading with our small sample), an unreasonable >3800 people would need to be recruited to detect anticipatory effects.

Effect size calculation is also stimulated to provide the magnitude of the main effects observed in the results.

> We have now added standardized effect sizes (standardized mean differences) for all of our pairwise comparisons.

Line #49: Considering that learning effect is an important topic of this study, the authors should explain or refer to a protocol already published in literature that tested if learning effect is minmized after only one week.

> We now made reference to what we could find in the existing literature on learning effects in our discussion. However, there is relatively little research investigating learning effects with the Flanker. By randomizing the order of sessions and controlling for session number as well as session type, however, we adjust for the impact of learning effects across sessions. 

Line #153: It should be informed whether the use of several criteria to analyze the parameters of cognitive task is arbitrarily proposed by authors or is already in the literature. If so, please cite this original work.

>Thank you for this suggestion, we have added an explanation for why chose to use an integrated speed-accuracy score vs the more typical speed score. Though the measure we used is relatively new, it has already been used a fair amount within the literature. We have added a few example references that use it in the domain of cognition.

Results

The results are too much descriptive and data are repeated at some points data.

>We have reworked our results section and removed the repeated data.

Line #184: Regarding missing data of some participants concerning cognitive task parameters, were the remaining data considered for analysis or were the participants excluded? Was there any criteria for this decision?

>All remaining data was considered for analysis, that is, no data that we possessed was excluded. This is because things like pairwise deletion will often result in biased estimates (as the mechanism of missing data is far more likely to be Missing at Random rather than Missing Completely at Random) and was the main reason for our use of linear mixed effects models as opposed to a standard repeated measures ANOVA. We have reworded the sentence to make it more clear that all data was used in the analysis.

Line #189: The data could be reported according to relative values (i.e. %HRmáx) to increase the individuality of measure. The proximity of anaerobic thresold influences their cognitive control response, for example.

>This is a good point and so we’ve now reported all heart rate data as %HR max.

.Discussion

The discussion is superficial. A very simple relationship is made with the findings of the literature.

An important finding, the "learning effect" is poorly explorated. The authors can explore and relate these data with literature in conclusion, not just in the future directions.

> The learning effects were certainly of interest to us. It surprised us that there was almost no mention of learning effects with reference to the Flanker task in the literature, despite an obvious learning effect across sessions. This has import implications for future studies, where learning effects should either be minimized (difficult in a restricted time frame) or adjusted for within analyses. We have added a more thorough discussion. 

Line #291: Here the authors need to reinforce that catecholamines and cortisol were not assessed in the present study.

>We have added to the sentence to reinforce this.

Lines #291 and 298: unpadronized citation. These citations do not follow the journal's instructions

>Thanks for catching this, we have now fixed them.

Line #316: the term 'strong' appears to be inappropriately used. The authors reinforce multiple limitations of the present study. Also, they did not measure the magnitude of statistical difference (i.e. effect size).

>This is a great point, ‘strong’ is often used in relation to magnitude of effects so we have removed the word ‘strong’. We have also added effect sizes to our results. 

 

Reviewer #2: The authors presented a well-written manuscript on the effects of moderate exercise, and its anticipation effect on cognitive control. The authors also used a relatively new measure of the Flanker test.

The results are interesting and bring novelty to the literature. However, some methodological problems were found.

Major comments:

The authors adopted an even distribution of congruent and incongruent trials. However, Lehle and Hubner [1] pointed out the need to adopt a high frequency of congruent stimulus compared with incongruent, once with an even distribution, the participants can adapt to them. In that way, we cannot know if anticipation exercise did not affect cognition due to the low difficulty level of the task.

>Indeed, a higher frequency of congruent trials will increase the Flanker Congruency Effect. It is possible that Flanker parameters that elicit an increased Flanker Congruency Effect may be more sensitive to anticipatory effects, though that is not certain. In the case of exercise-related literature, use of an even distribution of congruent and incongruent trials has been common (recent examples: (1–5)), as we did here. Also, positive effects of exercise have also been observed in choice response time tasks. However, it remains possible that anticipation of exercise may be more strongly observed in very difficult tasks (which are more likely to suffer with stress) and we have added this point to our discussion.

The sample was composed of 55% of females. The literature indicates that cognitive control is modulated by estrogen levels [2-4]. However, the authors did not control participants’ menstrual cycle, and we did not know at which extend this can be influencing the results.

> We acknowledge the reviewers point that the female participants’ Flanker performance may have been influenced by their menstrual cycle. However, given the randomization across conditions, it seems unlikely that this would result in a systematic bias across conditions, though it likely adds noise.

Minor comments:

Line 40: The authors introduce the paragraph talking about exercise benefits to cognition. Then, they talk about the physiological and metabolic benefits of exercise. I believe this second sentence is not necessary.

>We agree and have deleted this sentence and rewrote the surrounding sentences.

Line 110: It seems that the sentence has a typing error.

>Thanks for catching this. Upon revisiting, the sentence did seem slightly confusing, so we have changed it.

Line 130: The author affirmed that they recorded the resistance during exercise condition, but they did not report these data in the results section.

>Thank you for catching this, Reviewer 1 mentioned this as well. We have now added it to the section 3.2 Exercise Characteristics.

Table 1. It not common to use this grid format in tables. I believe, for future submissions, the authors should fix it.

>We have changed the styling. 

1. Lehle C, Hubner R. On-the-fly adaptation of selectivity in the flanker task. Psychonomic Bulletin & Review 2008;15(4):814-8

2. Colzato LS, Hertsig G, van den Wildenberg WP, Hommel B. Estrogen modulates inhibitory control in healthy human females: evidence from the stop-signal paradigm. Neuroscience 2010;167(3):709-15 doi: 10.1016/j.neuroscience.2010.02.029[published Online First: Epub Date]|.

3. Hodgetts S, Weis S, Hausmann M. Estradiol-related variations in top-down and bottom-up processes of cerebral lateralization. Neuropsychology 2017;31(3):319-27 doi: 10.1037/neu0000338[published Online First: Epub Date]|.

4. Jacobs E, D'Esposito M. Estrogen shapes dopamine-dependent cognitive processes: implications for women's health. J Neurosci 2011;31(14):5286-93 doi: 10.1523/jneurosci.6394-10.2011[published Online First: Epub Date]|.

 

References

1. Lefferts WK, DeBlois JP, White CN, Heffernan KS. Effects of Acute Aerobic Exercise on Cognition and Constructs of Decision-Making in Adults With and Without Hypertension. Front Aging Neurosci. 2019; 

2. Won J, Alfini AJ, Weiss LR, Callow DD, Smith JC. Brain activation during executive control after acute exercise in older adults. Int J Psychophysiol. 2019; 

3. Du Rietz E, Barker AR, Michelini G, Rommel AS, Vainieri I, Asherson P, et al. Beneficial effects of acute high-intensity exercise on electrophysiological indices of attention processes in young adult men. Behav Brain Res. 2019; 

4. Tsai CL, Ukropec J, Ukropcová B, Pai MC. An acute bout of aerobic or strength exercise specifically modifies circulating exerkine levels and neurocognitive functions in elderly individuals with mild cognitive impairment. NeuroImage Clin. 2018; 

5. Beyer KB, Sage MD, Staines WR, Middleton LE, McIlroy WE. A single aerobic exercise session accelerates movement execution but not central processing. Neuroscience. 2017; 

6. Vandierendonck A. Further Tests of the Utility of Integrated Speed-Accuracy Measures in Task Switching. J Cogn. 2018;

---

## [Decision Letter · Decision Letter 1]

26 Oct 2020

PONE-D-20-21829R1

Moderate aerobic exercise, but not anticipation of exercise, improves cognitive control

PLOS ONE

Dear Dr. Maximilian Bergelt,

Thank you for submitting your manuscript to PLOS ONE. After careful consideration, we feel that it has merit but does not fully meet PLOS ONE’s publication criteria as it currently stands. Therefore, we invite you to submit a revised version of the manuscript that addresses the points raised during the review process.

The academic editor and the two reviewers believe that the revised manuscript has merit and that it has been dramatically improved compared to the previous version submitted to PLOS One. Indeed, the authors were highly responsive to all comments. Congratulations! Although both reviewers judged that the manuscript is acceptable for publication in the present form, I still think that the authors should explain how they have calculated the standardized mean differences (as required by the first reviewer) in the material and methods section. If the authors adequately address the issue mentioned above, this academic editor will handle the manuscript by himself and not send it again for external reviewer evaluation.

We look forward to receiving your revised manuscript.

Kind regards,

Samuel Penna Wanner, Ph.D.

Academic Editor

PLOS ONE

Additional Editor Comments (if provided):

Please consider:

1) To define the meaning of the abbreviation SMD and explain how the standardized mean differences were calculated. Please also indicate the threshold values used to classify the effects sizes based on the SMD calculation.

2) To present the heart rate data as absolute values and indicate the percentage of maximum heart rate for specific time points, particularly after the exercise.

The editor is looking forward to receiving a revised and improved version of the manuscript.

Reviewers' comments:

Reviewer's Responses to Questions

**Comments to the Author**

1. If the authors have adequately addressed your comments raised in a previous round of review and you feel that this manuscript is now acceptable for publication, you may indicate that here to bypass the “Comments to the Author” section, enter your conflict of interest statement in the “Confidential to Editor” section, and submit your "Accept" recommendation.

Reviewer #1: All comments have been addressed

Reviewer #2: All comments have been addressed

2. Is the manuscript technically sound, and do the data support the conclusions?

Reviewer #1: Yes

Reviewer #2: Yes

3. Has the statistical analysis been performed appropriately and rigorously? 

Reviewer #1: Yes

Reviewer #2: Yes

4. Have the authors made all data underlying the findings in their manuscript fully available?

Reviewer #1: Yes

Reviewer #2: Yes

5. Is the manuscript presented in an intelligible fashion and written in standard English?

Reviewer #1: Yes

Reviewer #2: Yes

6. Review Comments to the Author

Reviewer #1: (No Response)

Reviewer #2: (No Response)

7. PLOS authors have the option to publish the peer review history of their article (what does this mean?). If published, this will include your full peer review and any attached files.

Reviewer #1: **Yes: **João Gabriel Silveira-Rodrigues

Reviewer #2: **Yes: **Larissa Oliveira Faria

---

## [Author Response · Author response to Decision Letter 1]

27 Oct 2020

Rebuttal Letter

PONE-D-20-21829

Moderate aerobic exercise, but not anticipation of exercise, improves cognitive control

PLOS ONE

Dear Dr. Wanner,

Thank you for the opportunity to revise our manuscript as well as the thoughtful comments and suggestions. In our revised manuscript, we have addressed your additional comments with our responses detailed below. Alterations to the manuscript addressing your comments are highlighted in yellow.

Thank you again for your suggestions, we believe that they significantly improve the clarity of this manuscript.

Best regards,

Max Bergelt

Additional Editor Comments:

1) To define the meaning of the abbreviation SMD and explain how the standardized mean differences were calculated. Please also indicate the threshold values used to classify the effects sizes based on the SMD calculation.

>We have added the meaning of SMD and explained how it was calculated in section ‘2.5 Statistical analysis’. We have also added thresholds for small, medium, and large effects in order to clarify interpretation.

2) To present the heart rate data as absolute values and indicate the percentage of maximum heart rate for specific time points, particularly after the exercise.

>We now present both the absolute heart rates at each time point, as well as percentage of maximum heart rate. The absolute heart rate is presented in Table 1. The %HR max during and after the intervention are presented in the text of ‘3.2 Exercise characteristics’.

---

## [Editor Report · Decision Letter 2]

30 Oct 2020

Moderate aerobic exercise, but not anticipation of exercise, improves cognitive control

PONE-D-20-21829R2

Dear Dr. Maximilian Bergelt,

We’re pleased to inform you that your manuscript has been judged scientifically suitable for publication and will be formally accepted for publication once it meets all outstanding technical requirements.

Kind regards,

Samuel Penna Wanner, Ph.D.

Academic Editor

PLOS ONE

Additional Editor Comments (optional):

The authors have adequately addressed my comments. Thank you.
---

## [Editor Report · Acceptance letter]

5 Nov 2020

PONE-D-20-21829R2 

Moderate aerobic exercise, but not anticipation of exercise, improves cognitive control 

Dear Dr. Bergelt:

I'm pleased to inform you that your manuscript has been deemed suitable for publication in PLOS ONE. Congratulations! Your manuscript is now with our production department. 

Kind regards, 

on behalf of

Dr. Samuel Penna Wanner 

Academic Editor

PLOS ONE